# GLOMA: GLOBAL VIDEO TEXT SPOTTING WITH MORPHOLOGICAL ASSOCIATION

**Han Wang**
Bytedance

**Yanjie Wang**
Bytedance

**Yang Li**
Bytedance

**Can Huang**
Bytedance

## ABSTRACT

Video Text Spotting (VTS) is a fundamental visual task that aims to predict the trajectories and content of texts in a video. Previous works usually conduct local associations and apply IoU-based distance and complex post-processing procedures to boost performance, ignoring the abundant temporal information and the morphological characteristics in VTS. In this paper, we propose GLOMA to model the tracking problem as global associations and utilize the Gaussian Wasserstein distance to guide the morphological correlation between frames. Our main contributions can be summarized as three folds. 1). We propose a Transformer-based global tracking method GLOMA for VTS and associate multiple frames simultaneously. 2). We introduce a Wasserstein distance-based method to conduct positional associations between frames. 3). We conduct extensive experiments on public datasets. On the ICDAR2015 video dataset, GLOMA achieves **56.0** MOTA with **4.6** absolute improvement compared with the previous SOTA method and outperforms the previous Transformer-based method by a significant **8.3** MOTA.

## 1 INTRODUCTION

Video Text Spotting is an essential topic in computer vision, which facilitates video understanding, video retrieval, and video captioning. By simultaneously carrying out detection, recognition, and tracking, VTS can locate and recognize the texts in each frame and build trajectories through time. In Fig. 1, the expansion of the Multi-Object Tracking (MOT) framework, as seen in studies like Wu et al. (2022b); Koo & Kim (2013); Tian et al. (2016); Cheng et al. (2020); Gao et al. (2021); Feng et al. (2021); Yu et al. (2021), involves a bilateral matching approach across two consecutive frames. These studies leverage appearance and positional relations, with a notable emphasis on the IoU score for text matching and trajectory construction. In case of trajectory interruptions, they retain information solely from the final frame where the text is detected, in line with MOT practices. We note that text characteristics in videos differ from those of pedestrians, cars, or other common MOT objects. Text transformations evolve more slowly than those in MOT, with texts experiencing minimal deformation from limb or pose changes, suggesting a more gradual visual progression. This stability over time often results in more consistent text appearances, enabling the use of global features to counter issues like blur. Additionally, text frequently undergoes swift translations due to camera movements, leading to weaker positional associations compared to MOT objects. Nonetheless, text shapes usually remain more stable than MOT objects, given the static nature of texts themselves. These observations suggest the importance of relying on global information rather than solely depending on features from individual frames, and highlight the need to consider morphological details over positional relationships. Thus, how to explicitly utilize temporal information and properly conduct morphological correlation in text scenarios remains a question.

In this paper, we propose a novel model GLOMA with global associations to explicitly use temporal information and a shape-aware distance to measure morphological similarity. We modify the detector YOLOX Ge et al. (2021) to detect texts as polygons in each frame. The tracking embeddings are extracted by Rotated RoIAlign Liu et al. (2018) and supervised by recognition loss to obtain semantic awareness. To utilize temporal information, a global embedding pool is maintained during the whole inference process to hold the historical tracking embeddings and trajectory information. Then a Transformer-based architecture is proposed to access long-range temporal associations by conducting associations between texts in the current frame and texts in

the global embedding pool for each frame. We also introduce a Wasserstein distance-based Yang et al. (2021) method as the positional measurement, which takes both location and morphology into account.

To prove the effectiveness of the proposed method, we conduct extensive experiments on several datasets and achieve state-of-the-art performance. On ICDAR2015 video Karatzas et al. (2015) dataset, our GLOMA obtains **56.0** MOTA on the test split, with **4.6** absolute improvement compared with the previous SOTA method Wu et al. (2022b), and outperforms the previous Transformer-based method Wu et al. (2022a) by **8.3** MOTA. On the ICDAR2013 Karatzas et al. (2013) video and Minetto Minetto et al. (2011) datasets, our GLOMA also reaches leading performance. Our GLOMA can run at around 20 FPS and the global association procedure takes 3.6 ms per frame on a single Tesla V100 GPU.

## 2    RELATED WORK

### 2.1    SCENE TEXT DETECTION

Different from object detection, Scene Text Detection aims to detect arbitrarily shaped texts in images. Benefiting from the development of object detection, Tian et al. (2016); Zhong et al. succeed in horizontal text detection by adopting similar methods. Based on FCN Long et al. (2015), EAST Zhou et al. (2017) is proposed to detect texts with different angles. PSENet Wang et al. (2019a) and PAN/PAN++ Wang et al. (2019b; 2021) adopt kernel-based methods and apply post-processing procedures to produce the final detecting results.

### 2.2    MULTI-OBJECT TRACKING

Multi-Object Trackin Bewley et al. (2016); Wojke et al. (2017); Zhou et al. (2020); Zhang et al. (2021); Wang et al. (2020); Zhou et al. (2022b) aims to predict the coordinates of each object in each frame. Most existing methods Bewley et al. (2016); Wojke et al. (2017); Zhou et al. (2020); Zhang et al. (2021); Wang et al. (2020) model the tracking task as a bilateral matching problem between instances in two adjacent frames. Wojke et al. (2017) adopts separate detection and tracking networks and a tracking-by-detection pipeline, with an *IoU*-based positional distance and cascaded matching procedures. To simplify the pipeline, Wang et al. (2020); Zhang et al. (2021) introduce a joint-detection-and-tracking protocol, which combines both detection and tracking in a single network. How-

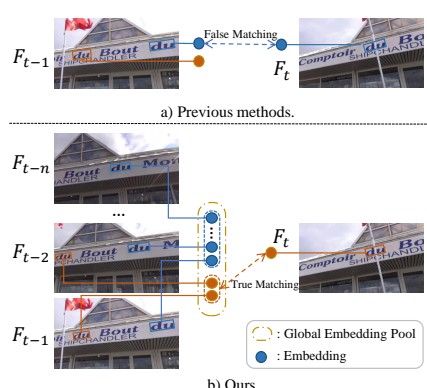

Figure 1: Motivation. Previous works usually conduct local associations and easily fail in scenes with interference (*e.g.,* identical texts). To solve the problems, we introduce global associations to utilize temporal information to make our method more robust towards such scenes.

ever, they highly rely on complex post-processing procedures and require many handcrafted hyperparameters. Recently, some works Sun et al. (2020); Zeng et al. (2021) model tracking as a query problem, treating different trajectories as different queries and decoding the corresponding coordinates with a Transformer-based architecture. Though a more precise pipeline, these approaches usually fail in crowded scenes due to the absence of explicit positional awareness.

### 2.3    VIDEO OBJECT DETECTION

Video Object Detection (VOD) aims to boost the detection performance by aggregating context features. Attention blocks are widely used in Chen et al. (2020); Wu et al. (2019); Deng et al. (2019); Shvets et al. (2019); Zhou et al. (2022a); Wang et al. (2022); Zhu et al. (2017) to conduct correlations between reference images and the current image, achieving awareness of long-range temporal information. Based on a two-stage detector, Chen et al. (2020); Wu et al. (2019); Deng et al. (2019); Shvets et al. (2019); Zhu et al. (2017) aggregate features after RoIs Girshick (2015);

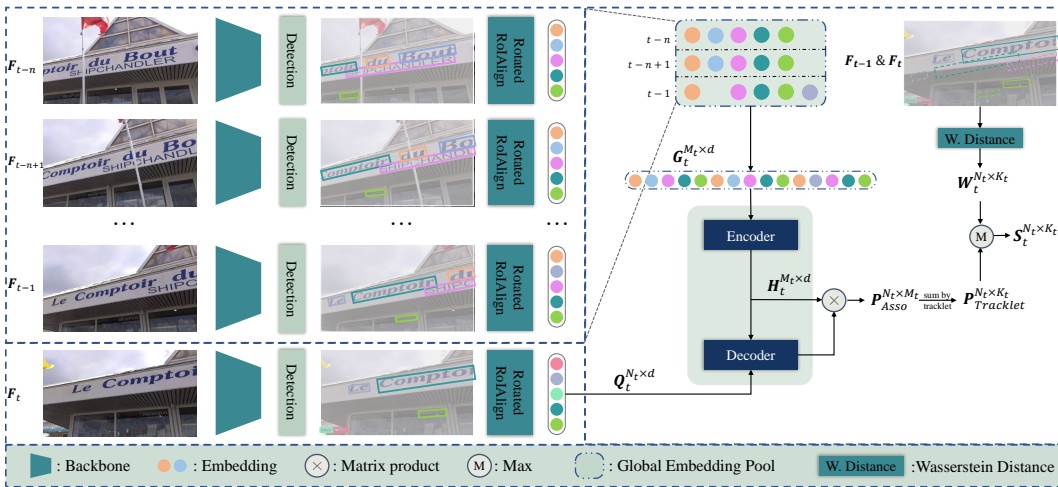

Figure 2: Overview. A global embedding pool is maintained to store historical tracking embeddings and trajectory information and is updated after each frame. With a shallow Transformer layer, we conduct associations between embeddings of the current frame and embeddings in the global embedding pool to obtain the global association score. Furthermore, a Wasserstein distance-based method is applied to measure the positional similarity between texts in frames. Some detailed architectures are ignored for clarity.

He et al. (2017) to gain enhanced features. In particular, Chen et al. (2020) designs a hierarchical structure to aggregate local and global features. Based on Transformer, Wang et al. (2022); Zhou et al. (2022a) aggregate queries through time, also achieving temporal awareness.

### 2.4 VIDEO TEXT TRACKING AND VIDEO TEXT SPOTTING

Given a video clip, Video Text Tracking (VTT) aims to predict the coordinates of each text in each frame and Video Text Spotting further requires recognition results. Existing methods Li et al. (2021); Wu et al. (2022b;a; 2021); Cheng et al. (2020); Feng et al. (2021) succeed in most common scenes by conducting local associations. For example, a typical structure consists of a backbone, a detector and an RoI to extract instance-level features. The appearance similarity between instances is measured by a pairwise distance (*e.g.,* cosine distance). The positional association score is calculated by the *IoU* between instances in frames. A cascaded post-processing is applied to fully use the appearance similarity and positional association. Motivated by Zeng et al. (2021); Sun et al. (2020), some models Wu et al. (2022a; 2021) directly apply Transformer-based architectures to VTS, with an extra network for recognition. However, lacking *IoU*-based post-processing procedures and utilization of temporal information, these methods struggle in many difficult VTS situations (*e.g.,* crowded scenes, fast movement, lighting change).

## 3 METHODS

### 3.1 OVERVIEW

GLOMA is an end-to-end framework for Video Text Spotting, which conducts global associations and adopts morphology-aware measurements. The whole framework can be seen in Fig. 2. There are three parallel heads: detection head, recognition head, and tracking head. Given a video, we first detect the potential objects as 4-point coordinates in Frame $F_t$, and then extract corresponding tracking embeddings as $e_t^i$ for each object. We maintain a global embedding pool as $\mathbb{G}_t$, and the concatenated features of all the embeddings in $\mathbb{G}_t$ are represented as $\boldsymbol{G}_t \in \mathbb{R}^{M_t \times d}$, where $M_t = \sum_{i=t-L}^{t-1} N_i$ stands for the number of embeddings in $\mathbb{G}_t$. $N_t$ is the number of texts in frame $F_t$, $d$ is the dimension of each embedding, and $L$ is the sliding window size. Our tracking head calculates association scores between objects in frame $F_t$ and objects in $\mathbb{G}_t$, generating an association matrix

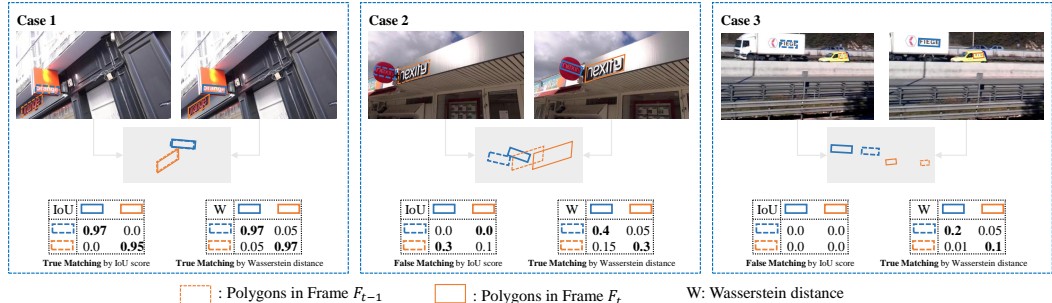

Figure 3: Three cases to demonstrate the effectiveness of Wasserstein distance. *IoU*-based distance and Wasserstein distance both succeed in Case 1. But in Case 2 and Case 3, the fast movements result in the poor performance of *IoU*-based distance, where Wasserstein distance produces more steady results by considering both location and morphology.

$\boldsymbol{P}_{Asso} \in \mathbb{R}^{N_t \times M_t}$, and then $\boldsymbol{P}_{Asso}$ is turned into a tracklet-level association matrix represented as $\boldsymbol{P}_{Tracklet} \in \mathbb{R}^{N_t \times K_t}$, where $K_t$ is the number of tracklets in the global embedding pool $\mathbb{G}_t$. Besides, we also calculate the morphological similarities between the two adjacent frames $F_{T-1}$ and $F_T$ and output a distance matrix $\boldsymbol{W}_t \in \mathbb{R}^{N_t \times K_t}$. The two score matrices $\boldsymbol{P}_{Tracklet}$ and $\boldsymbol{W}_t$ are united by a simple max operation without any post-processing and output the final scores depicted as $S_t \in \mathbb{R}^{N_t \times K_t}$ in Fig. 2. And a Hungarian algorithm is applied to assign IDs.

## 3.2 GLOBAL TRACKING

We adopt a Transformer-based network for global tracking similar to Zhou et al. (2022b). All the previous embeddings in the global embedding pool are encoded by a Transformer encoder as global memories. With an extraordinary long-range temporal modeling ability, Transformer is able to capture global information. Transformer decoder inputs encoded historical information as memory, with embeddings in the current frame as queries to calculate the similarity scores between the current instances and trajectories. The whole procedure can be written as:

$$\boldsymbol{H}_t = \text{Encoder}(\boldsymbol{G}_t), \tag{1}$$

$$\boldsymbol{P}_{Asso} = \text{Decoder}(\boldsymbol{Q}_t, \boldsymbol{H}_t)\boldsymbol{H}_t^T, \tag{2}$$

where $\boldsymbol{H}_t \in \mathbb{R}^{M_t \times d}$ denotes the encoded historical temporal memory. $\boldsymbol{P}_{Asso} \in \mathbb{R}^{N_t \times M_t}$ is the output association matrix. $\boldsymbol{Q}_t \in \mathbb{R}^{N_t \times d}$ refers to the query embeddings (*i.e.*, embeddings of the current frame).

During training, we associate all the embeddings with themselves (*i.e.*, $\boldsymbol{Q}_t = \boldsymbol{G}_t$), generating an association matrix $\boldsymbol{A} \in \mathbb{R}^{M \times M}$, where $M = \sum_{t=1}^{B} N_t$ is the number of all the texts within a batch and $B$ represents the number of images in a batch, which is fixed as 16 in our experiments. For each text in each timestamp $t$, we have a vector $\boldsymbol{a} \in \mathbb{R}^{N_t+1}$ which indicates the association scores between one query embedding $\boldsymbol{q}_i$ and embeddings in frame $t$. Note that the extra dimension in $\boldsymbol{a}$ refers to the empty association (*i.e.*, the query has no matched target in this frame, usually indicating an occlusion or the end of the trajectory). A softmax function is used to transform the score into the possibility $\boldsymbol{P}_{Asso}$:

$$\boldsymbol{P}_{Asso}(\boldsymbol{q}_i, \boldsymbol{e}_j) = \frac{\exp(\boldsymbol{a}_j)}{\sum_{j \in \{\emptyset, 1, \dots N_t\}} \exp(\boldsymbol{a}_j)}. \tag{3}$$

Thus, we learn the scores by minimizing the log-likelihood of each possibility. For each tracklet set $\mathbb{T}_k, k = 1, 2, \dots, K$, we assume there are $N_k$ embeddings in $\mathbb{T}_k$ and for each query embedding $\boldsymbol{e}_i$ in the global embedding pool $\mathbb{G}$, we calculate the loss only when $\boldsymbol{e}_i \in \mathbb{T}_k$. The loss writes as:

$$\ell_{tracklet}(\mathbb{T}_k, \boldsymbol{q}_i) = -\sum_{j=1}^{N_k} \mathbb{1}_{\boldsymbol{q}_i \in \mathbb{T}_k} \cdot \log \boldsymbol{P}_{Asso}(\boldsymbol{q}_i, \boldsymbol{e}_j),$$

$$\ell_{track} = \sum_{k=1}^{K} \sum_{i=1}^{M} \ell_{tracklet}(\mathbb{T}_k, \boldsymbol{q}_i), \tag{4}$$

where $1_{q_i \in \mathbb{T}_k}$ is 1 if the query embedding belongs to the tracklet set $\mathbb{T}_k$.

**Semantic embeddings.** To increase the discrimination of embeddings, we also introduce semantic information to boost the performance of tracking. In detail, the embeddings fed into Transformer are extracted by Rotated RoIAlign with a shallow convolutional layer and an LSTM Hochreiter & Schmidhuber (1997) layer, followed by a fully connected layer to project features into the classes of words. The architecture writes as:

$$
\begin{aligned}
\boldsymbol{e}_t &= \mathrm{lstm}(\mathrm{conv}(\mathrm{r\text{-}roi}(\boldsymbol{X}_t))), \\
\boldsymbol{o}_t &= \mathrm{fc}(\boldsymbol{e}_t),
\end{aligned}
\tag{5}
$$

where $\boldsymbol{X}_t$ is the backbone feature map in frame $F_t$, $\boldsymbol{e}_t$ is the embedding fed into Transformer for associations, and $\boldsymbol{o}_t$ is the recognition output supervised by Connectionist Temporal Classification (CTC) Graves et al. (2006) loss.

### 3.3 WASSERSTEIN DISTANCES IN CORRELATION

Concerning morphological information, we apply the Wasserstein distance to model both location similarity and shape similarity. Previous methods usually measure the location similarity between two adjacent frames by a pairwise calculation of *IoU* of each pair of bounding boxes and ignore the shape similarity, just the same as the methods in MOT. However, considering the differences from the scenes in MOT, the shapes of the texts are much more steady in a small time window, which can also be a strong feature for tracking.

As demonstrated in Fig. 3, we use three cases to exhibit the advantage of Wasserstein distance over the *IoU*. In Case 1, all texts are moving at a low speed, so the location information is enough for associations, and both *IoU* and Wasserstein distance can conduct a correct match. In Case 2, the fast movement of the camera leads to the fast drift of texts. In this situation, the *IoU* scores give a false positional association clue, leading to a false matching. However, the morphological differences are obvious so that the Wasserstein distance can capture the incoherence. In Case 3, the *IoU* scores are both 0 due to the fast movement of cars, while the Wasserstein distance can still perform the correct positional association.

To obtain the awareness of both the locations and shapes, we model the polygons in different frames as Gaussian distributions and measure the similarity via distribution distance. For each predicted 4-point coordinate $b$ in two adjacent frames, we calculate pairwise Wasserstein distances between the corresponding convex hull rotated bounding boxes $\boldsymbol{b}(x, y, w, h, \theta)$. The first step is to convert the rotated box $b$ into Gaussian distribution $\mathcal{N}(\boldsymbol{\mu}, \boldsymbol{\sigma})$:

$$
\begin{aligned}
\boldsymbol{\mu} &= (x, y), \\
\boldsymbol{\sigma} &= \mathbf{R}\mathbf{S}\mathbf{R}^\top \\
&= \begin{pmatrix} \cos\theta & -\sin\theta \\ \sin\theta & \cos\theta \end{pmatrix} \begin{pmatrix} \frac{w}{2} & 0 \\ 0 & \frac{h}{2} \end{pmatrix} \begin{pmatrix} \cos\theta & \sin\theta \\ -\sin\theta & \cos\theta \end{pmatrix} \\
&= \begin{pmatrix} \frac{w}{2}\cos^2\theta + \frac{h}{2}\sin^2\theta & \frac{w-h}{2}\cos\theta\sin\theta \\ \frac{w-h}{2}\cos\theta\sin\theta & \frac{w}{2}\sin^2\theta + \frac{h}{2}\cos^2\theta \end{pmatrix},
\end{aligned}
\tag{6}
$$

where $\mathbf{R}$ is the rotated matrix and $\mathbf{S}$ is the diagonal matrix. The Wasserstein distance between two Gaussian distributions is represented as:

$$
d^2 = \|\boldsymbol{\mu}_1 - \boldsymbol{\mu}_2\|_2^2 + \mathrm{Tr}\left(\boldsymbol{\sigma}_1 + \boldsymbol{\sigma}_2 - 2\left(\boldsymbol{\sigma}_1^{1/2}\boldsymbol{\sigma}_2\boldsymbol{\sigma}_1^{1/2}\right)^{1/2}\right).
\tag{7}
$$

With proper consideration of both the angles and the coordinates, Wasserstein distance can capture both location similarity and morphological similarity. Finally, to convert the distance into an applicable positional score, we have:

$$
\begin{aligned}
\boldsymbol{W}(b_1, b_2) &= \boldsymbol{W}\left(\mathcal{N}(\boldsymbol{\mu}_1, \boldsymbol{\sigma}_1); \mathcal{N}(\boldsymbol{\mu}_2, \boldsymbol{\sigma}_2)\right), \\
&= 1 - \frac{\alpha d}{f(\boldsymbol{\sigma}_1, \boldsymbol{\sigma}_2)},
\end{aligned}
\tag{8}
$$

where $\alpha$ is a hyper-parameter, and $f$ is a function to normalize the distance. We set $f(\sigma_1, \sigma_2) = (\mathrm{Tr}(\sigma_1\sigma_2))^{1/4}$.

### 3.4 Loss functions

There are three tasks and three corresponding losses. For the detection head, we adopt L1 loss to regress the 4-point polygons and other losses are set the same as losses in YOLOX Ge et al. (2021). For the recognition head, we adopt Connectionist Temporal Classification (CTC) Graves et al. (2006) loss for texts. We also apply multi-task learning losses. The whole losses are written as:

$$\ell = e^{-\sigma_1}\ell_{\det} + e^{-\sigma_2}\ell_{rec} + e^{-\sigma_3}\ell_{track} + \sigma_1 + \sigma_2 + \sigma_3, \tag{9}$$

where $\sigma_1, \sigma_2, \sigma_3$ are learnable parameters.

### 3.5 Inference

During inference, we iteratively build the trajectories. For the initial frame $F_0$, we regard each text as the start of a trajectory. For frame $F_t$, we have a global embedding pool $\mathbb{G}_t$ which stores all the previous embeddings in a sliding window, and the corresponding embedding matrix writes as $\boldsymbol{G}_t$ (*i.e.,* the concatenated embeddings). Assume there are $K_t$ trajectories in $\mathbb{G}_t$, and each trajectory has $N_k$ embeddings. We regard embeddings in the current frame $F_t$ as $\mathbb{Q}_t$ and the corresponding embedding matrix as $\boldsymbol{Q}_t$. The association score $A_t$ is calculated between $\boldsymbol{Q}_t$ and $\boldsymbol{G}_t$, and converted to a possibility matrix $\boldsymbol{P}_{Asso}$ by a softmax function. Then a tracklet-wise sum is applied to calculate the possibility of each tracklet. For each $\boldsymbol{q}_i \in \mathbb{Q}_t$ we have:

$$\boldsymbol{P}_{Tracklet}(\boldsymbol{q}_i, \boldsymbol{G}_t) = \sum_{j=1}^{N_k} \boldsymbol{P}_{Asso}(\boldsymbol{q}_i, \boldsymbol{e}_j), \tag{10}$$

where $\boldsymbol{e}_j$ is the embedding in each trajectory. After calculating all the embeddings in $\mathbb{Q}_t$, we can finally obtain a matrix $\boldsymbol{P}_{Tracklet} \in \mathbb{R}^{N_t \times K_t}$, which refers to the possibility that the embedding belongs to the tracklet. Also, Wasserstein distance between frame $F_{t-1}$ and $F_t$ is calculated as a positional and morphological similarity score $\boldsymbol{W}_t$, and the final output is $\max(\boldsymbol{P}_{Tracklet}, \boldsymbol{W}_t)$. A Hungarian algorithm is applied to ensure the ID assignment is unique for each text.

## 4 Experiments

### 4.1 Implemented details

We adopt ResNet-50 He et al. (2016) with FPN Lin et al. (2017) layers as our backbone and use the checkpoint pretrained on ImageNet Deng et al. (2009); Russakovsky et al. (2015). The architecture of the detection head is borrowed from YOLOX Ge et al. (2021) with an extra branch to regress the polygons. The tracking head is a lightweight architecture with only a one-layer Transformer. All experiments are conducted on Tesla V100 GPUs. We first pretrain the model on COCOText Veit et al. (2016) and apply Random Crop, Random Resize, Random Color Jittering, and Pseudo Track Zhou et al. (2020) for data augmentation. Then it is fine-tuned on other datasets. The batch size is fixed as 16 when training and random sampling within a clip is applied to make sure images in a batch are from the same video clip. During inference, we resize the images with the shorter side fixed and the ratio of images kept.

### 4.2 Datasets and metrics

Following previous protocols, we evaluate our methods on several different datasets.
**ICDAR2015 video and ICDAR2013 video**. ICDAR2015 video contains 25 clips for training and 24 clips for testing. Most scenes are street views with tens of texts in one image. ICDAR2013 video is a sub-dataset of ICDAR2015 video.
**Minetto**. Minetto is a small dataset that contains 5 videos harvested outdoors. Without a training split, it is used as a test dataset in previous methods.
**Metrics**. Following previous protocols Ristani et al. (2016), we adopt the metrics inherited from MOT. Different from MOT, metrics in Video Text Tracking adopt the *IoU* between polygons to measure the similarity of two instances. Three metrics, MOTA, MOTP, and IDF1, are mainly used to evaluate performance. MOTA measures a comprehensive performance of both the detection and tracking performance, and MOTP mainly concerns the ability to fit the bounding boxes. IDF1 only

Table 1: Video Text Tracking on different datasets. Our proposed method outperforms previous methods by a large margin.

| Dataset | Methods | MOTA↑ | MOTP↑ | IDF1↑ | MM↑ | ML↓ |
|---|---|---|---|---|---|---|
| ICDAR2015 video | AJOU Koo & Kim (2013) | 16.4 | 72.7 | 36.1 | 14.1 | 62.0 |
| | Free Cheng et al. (2020) | 43.2 | 76.7 | 57.9 | 36.6 | 44.4 |
| | SAVTD Feng et al. (2021) | 44.1 | 75.2 | 58.2 | 44.8 | 29.0 |
| | SVRep Li et al. (2021) | 49.5 | 73.9 | 66.1 | 44.9 | 27.1 |
| | CoText Wu et al. (2022b) | 51.4 | 73.6 | 68.6 | 49.6 | **23.5** |
| | TransVTSpotter Wu et al. (2021) | 44.1 | 75.8 | 57.3 | 34.3 | 33.7 |
| | TransDETR Wu et al. (2022a) | 47.7 | 74.1 | 65.5 | 42.0 | 32.1 |
| | Ours | **56.0** | **77.4** | **70.5** | **49.7** | 27.3 |
| ICDAR2013 video | YORO Cheng et al. (2019) | 47.3 | 73.7 | 62.5 | 33.1 | 45.3 |
| | SVRep Li et al. (2021) | 53.2 | 76.7 | 65.1 | 38.2 | 33.2 |
| | CoText Wu et al. (2022b) | 55.8 | 76.4 | 68.1 | 44.6 | 28.7 |
| | TransDETR Wu et al. (2022a) | 54.7 | 76.6 | 67.2 | 43.5 | 33.2 |
| | Ours | **56.3** | **78.7** | **68.6** | **46.0** | **28.6** |
| Minetto | SAVTD Feng et al. (2021) | 83.5 | 76.8 | - | - | - |
| | SVRep Li et al. (2021) | 86.3 | **81.0** | 83.9 | **96.4** | **0** |
| | CoText Wu et al. (2022b) | 86.9 | 80.6 | 83.9 | 87.7 | 0 |
| | TransVTSpotter Wu et al. (2021) | 84.1 | 77.6 | 74.7 | - | - |
| | TransDETR Wu et al. (2022a) | 84.1 | 57.9 | 76.7 | 36.6 | 44.4 |
| | Ours | **87.1** | 80.6 | **84.2** | 89.3 | 3.6 |

Table 2: Video Text Spotting on ICDAR2015 video dataset. Our GLOMA also achieves leading performance.

| Methods | MOTA↑ | MOTP↑ | IDF1↑ | MM↑ | ML↓ |
|---|---|---|---|---|---|
| Free Cheng et al. (2020) | 53.0 | 74.9 | 61.9 | 45.5 | 35.9 |
| CoText Wu et al. (2022b) | 59.0 | 74.5 | 72.0 | 48.6 | 26.4 |
| TransVTSpotter Wu et al. (2021) | 53.2 | 74.9 | 61.5 | - | - |
| TransDETR Wu et al. (2022a) | 58.4 | 75.2 | 70.4 | 32.0 | **20.8** |
| TransDETR (aug) Wu et al. (2022a) | 60.9 | 74.6 | 72.8 | 33.6 | 20.8 |
| OURS | **62.5** | **78.2** | **74.2** | **51.0** | 22.0 |

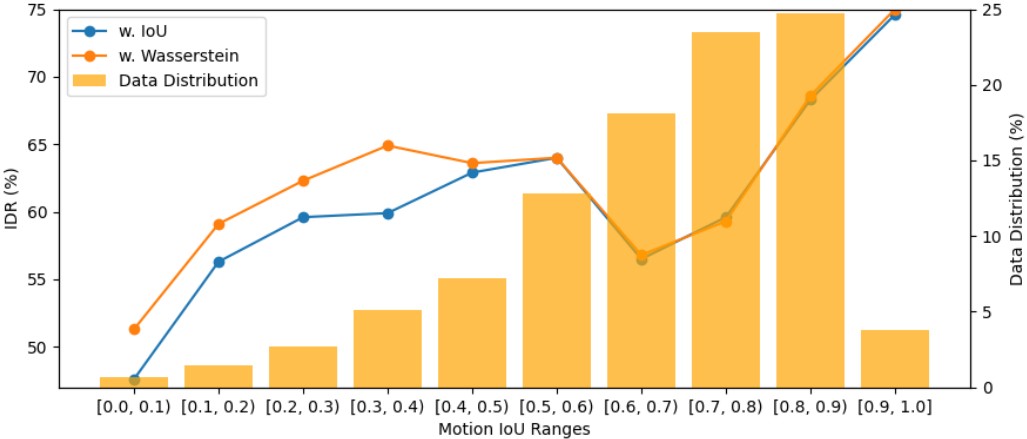

Figure 4: Motion-aware evaluations on the ICDAR13 video dataset

Table 3: Ablation study on the sliding window size. As the window size goes larger, the tracking performance tends to improve.

| Window size | MOTA↑ | MOTP↑ | IDF1↑ | MM↑ | ML↓ | Asso. T (ms)↓ |
|---|---|---|---|---|---|---|
| 2 | 55.1 | **77.5** | 63.0 | 46.8 | 30.1 | **3.1** |
| 4 | 55.9 | 77.4 | 68.3 | 49.4 | 28.2 | 3.2 |
| 8 | **56.0** | 77.4 | 70.5 | **49.7** | 27.3 | 3.6 |
| 16 | 55.7 | 77.4 | **71.0** | 49.1 | **27.1** | 3.9 |

Table 4: Experiments on positional distance (**Row 1-3**), self attention layer in decoder (**Row 4-5**), max operation (**Row 6-8**), and semantic embeddings (**Row 9-10**).

| Methods | MOTA↑ | MOTP↑ | IDF1↑ | MM↑ | ML↓ |
|---|---|---|---|---|---|
| *w/o.* distance | 55.5 | 77.4 | 70.0 | 47.3 | 28.1 |
| *w.* IoU | 55.8 | 77.4 | **70.6** | 49.0 | **27.1** |
| *w.* Wasserstein | **56.0** | **77.4** | 70.5 | **49.7** | 27.3 |
| *w.* self-attn | 53.2 | 77.1 | 69.8 | **52.8** | **25.4** |
| *w/o.* self-attn | **56.0** | **77.4** | **70.5** | 49.7 | 27.3 |
| *w.* $\boldsymbol{P}_{Tracklet}$ | 55.5 | 77.4 | 70.0 | 47.3 | 28.1 |
| *w.* $\boldsymbol{W}_t$ | 43.2 | **77.8** | 50.2 | 33.0 | 40.2 |
| *w.* $max(\boldsymbol{P}, \boldsymbol{W})$ | **56.0** | 77.4 | **70.5** | **49.7** | **27.3** |
| *w/o.* semantic | 53.9 | 77.4 | 66.9 | 47.7 | 28.0 |
| *w.* semantic | **56.0** | **77.4** | **70.5** | **49.7** | **27.3** |

measures the ability of tracking. Besides, we also adopt Mostly-Matched (MM) and Mostly-Lost (ML) to evaluate the completeness of trajectories. In Video Text Spotting, we also use the same metrics to measure the performance, but the similarity between instances is calculated by the edit distance between texts.

## 4.3 STATE-OF-THE-ART COMPARISONS

To verify the effectiveness of the proposed GLOMA, we compare its performance with several SOTA methods.

**Video Text Tracking**. Video Text Tracking is a core task to measure the performance of all methods. Thus, we evaluate our method in VTT and compare with other methods on several public datasets. On the most commonly used dataset ICDAR2015 video, we outperform previous works by a large margin. We obtain 4.6 absolute improvement over the previous state-of-the-art method and 8.3 absolute improvement over the previous Transformer-based method, which proves the effectiveness of the proposed GLOMA. Besides, our method also achieves leading performance on MOTP and IDF1. On ICDAR2013 video, our GLOMA achieves top performance on all metrics. On a smaller dataset Minetto, we also achieve SOTA results.

**Video Text Spotting**. Video Text Spotting concerns the tracking performance and the recognition results. As shown in Tab. 2, compared with the previous SOTA method TransDETR (aug), Wu et al. (2022a) our GLOMA obtains 1.6 absolute improvement on MOTA, 3.6 absolute improvement on MOTP, and 1.4 absolute improvement on IDF1. Therefore, with a simple and shallow recognition head, our GLOMA can also achieve great performance.

## 4.4 ABLATION STUDIES

To verify the effectiveness of the components of GLOMA, we conduct several ablation studies on ICDAR2015 video, as shown in Tab. 3-4.

**Sliding window size**. We study how the size of the sliding window impacts the final results during

the inference stage. As shown in Tab. 3, we can see a trend that the tracking performance goes up when the window length increases. When the window length is 2, the situation only involves associations between the two adjacent frames (*i.e.,* local associations), and the performance is much worse than when using a larger window size. Note that the metric MOTA concerns both the detection performance and tracking performance, and IDF1 only focuses on the tracking performance. Besides the temporal awareness obtained in training process, a longer window during inference stage also brings longer range of temporal information, leading to a sharp increase in IDF1, which proves the contribution of global embeddings to the final tracking performance. Therefore, we adopt 8 as the default sliding window size.

**Distance measurement.** We adopt Wasserstein distance for the measurement of positional association instead of *IoU*. As mentioned above, Wasserstein distance pays attention to both the location and morphological features so that it outperforms *IoU* when applied as a positional distance measurement in VTS. As shown in Tab. 4 (Row 1-3), positional distance can improve the performance, and with Wasserstein distance, the proposed GLOMA can achieve better results than with *IoU*. We also conduct motion-aware evaluation similar to that in Video Object Detection (VOD) using motion IoU Zhu et al. (2017) to indicate the motion speed of texts. We evaluate GloTSFormer on the ICDAR13 video dataset, as the test labels of ICDAR15 video are on the webdriver which are unavailable. As we evaluate the performance on data segments with different motion IoUs, we use ID Recall (IDR) Ristani et al. (2016) instead of other metrics involving false positives because TPs of motion IoU range of $[0.3, 0.4)$ can be taken as FPs of $[0.4, 0.5)$. As shown in Fig. 4 (Overall IDR: 62.5% v.s. 62.9%), where W. refers to Wasserstein distance, when the motion IoU decreases (indicating faster motion), Wasserstein distance brings out greater improvements.

**Attention layer.** The encoder layer is one multi-head attention layer and the decoder layer includes one cross-attention layer without a self-attention layer. As shown in Tab. 4 (Row 4-5), we do not observe obvious improvement of the overall performance when introducing self-attention layer and we remove it.

**Max operation.** We show extra experiments in Tab. 4 (Row 8-10). Both $\boldsymbol{P}_{Tracklet}$ and $\boldsymbol{W}_t$ contribute to the performance.

**Semantic embeddings**. We use the embeddings before the final fully-connected layer for associations. The embeddings we select carry semantic information and can provide some prior information and boost tracking performance. To verify the effectiveness, we only feed Transformer the features after Rotated RoIAlign to explore the influence of semantic information. As shown in Tab. 4 (Row 11-12), the ablation studies show the effectiveness of the proposed GLOMA.

**Speed analysis.** Our GLOMA runs at around 20 FPS on a single Tesla V100 GPU. The backbone, FPN layers, detection head, and recognition head take about 48 ms per frame, and the tracking procedure takes about 3.6 ms per frame. By maintaining a global embedding pool, we do not have to repeatedly extract the embeddings from each frame. As shown in Tab. 3, Asso. T refers to the time cost by the tracking procedure (*i.e.*, the inference of Transformer and associations). When the window size goes larger, we only observe a slight increment in the time consumed.

## 4.5 VISUALIZATION

As demonstrated in Fig. 5, we select two videos to show the advantage of our GLOMA. From an overall perspective, our GLOMA performs fewer false assignments in tracking and fewer FNs in detection. As presented in video (a), when facing crowded scenes with motion blur, TransVTSpotter and TransDETR fail to perform the right assignments due to excessive interfering texts. We could notice that the polygon colors of the texts (*e.g.,* "CV", "de", "super", "Tel") are changed over time, indicating many ID switches. On the contrary, our GLOMA has a much more steady performance without any ID switch. As presented in video (b), the architecture of TransDETR and TransVTSpotter have some limitations in detection, resulting in False Negatives in some frames. Without a global view, TransVTSpotter and TransDETR tend to conduct incorrect assignments especially when the fractures of trajectories take place. Our detection performance is relatively independent of tracking, and our GLOMA has a global view of historical information, leading to better performance on both detection and tracking.

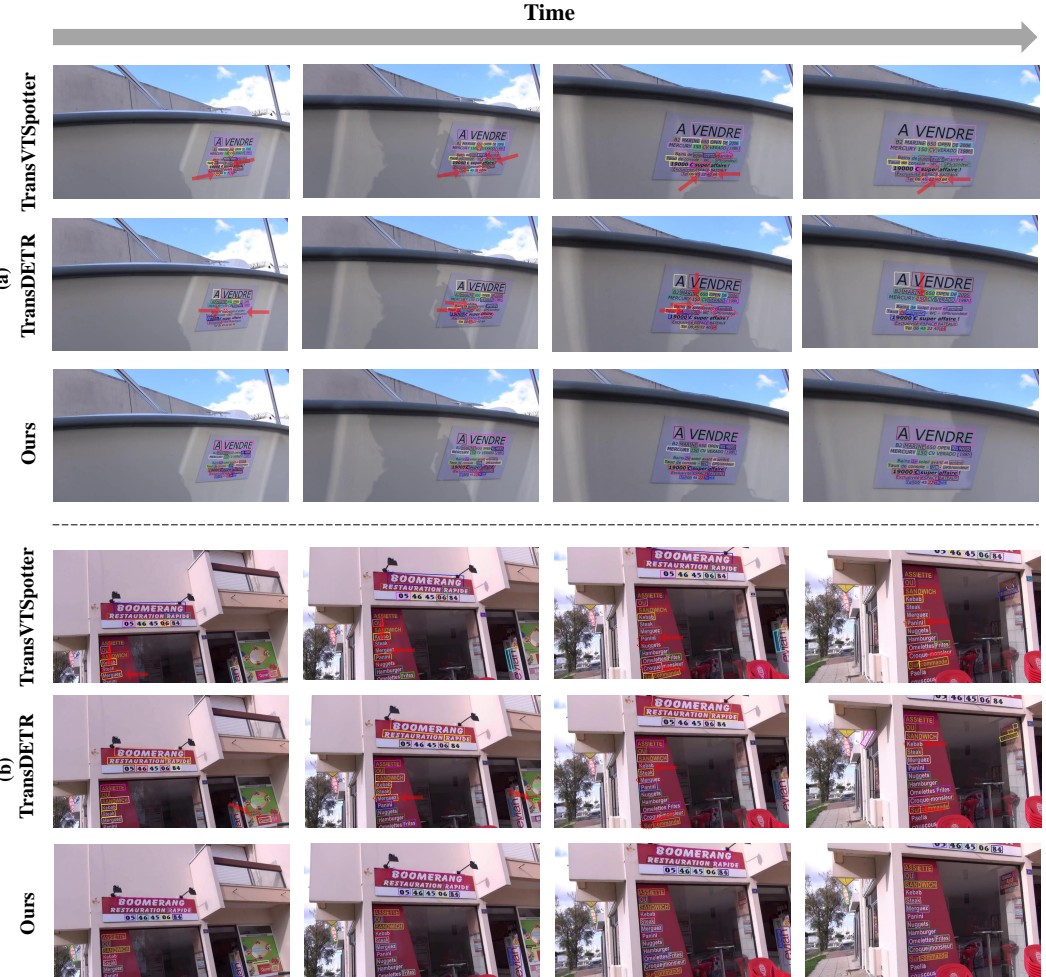

Figure 5: We demonstrate the results of previous Transformer-based methods Wu et al. (2021; 2022a) and our GLOMA. Different IDs are represented in different colors. Some of the false results (*e.g.,* FNs, ID switches, and IDFs) are marked with a dotted red circle and pointed out by a red arrow. Apparently, our GLOMA performs better than previous Transformer-based methods, especially in crowded scenes.

## 5 LIMITATIONS AND CONCLUSION

**Limitations.** Though GLOMA achieves great performance on VTS, we find the detector fails in detecting texts when facing severe motion blur and extreme sizes, which leads to fractures of trajectories. Our detector is less robust towards deterioration compared with our tracking method. We would consider how to apply trajectory information to improve detection performance in our future work.

**Conclusion.** In this work, we propose a Transformer-based global text spotting method GLOMA. We explore how to fully exploit temporal information and morphological information with a global association module and Wasserstein distance, respectively. We also conduct extensive experiments on the public datasets to verify the effectiveness of our method. We hope our work can shed light on future research on the utilization of temporal information and morphological information in Video Text Spotting.

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
