# OpenReview forum: "GLOMA: Global Video Text Spotting with Morphological Association"
_ICLR.cc/2025/Conference — ICLR 2025 Poster_

### Official Review · Reviewer_1EQ7 · 2024-10-29

**Soundness:** 2
**Presentation:** 1
**Contribution:** 2
**Rating:** 6
**Confidence:** 4

**Summary:**

This paper studies the video text spotting problem by introducing a GLOMA method as a new solution. The proposed GLOMA formulates a global association task to tackle the VTS problem, in which a Transformer tracking method and a Wasserstein distance method facilitates the tracking.

**Strengths:**

Writing looks good. Results are promising.

**Weaknesses:**

I have many concerns about the demonstration of novelty. The contributions of this paper also need better clarification. See the questions below.

**Questions:**

(1) After reading through the paper, it remains unclear to me why the authors need to devise a specific Transformer and specific association mechanism to fulfill the VTS. To my understanding, there are many SOTA multi-object tracking algorithms and single object tracking algorithms that seem perfectly fit the proposed task. The current SOTA methods do not simply rely on the IoU matching, despite that the VTS fields may have many IoU-based approaches. I am not saying that devising new approaches for VTS is not important, but at least the paper requires a much clearer justification of why existing SOTA multi-/single-object tracking algorithms cannot tackle this task effectively. The potentially related SOTA tracking methods include but not limited to [1,2,3,4]. The comparison (or at least discussion) about more SOTA tracking methods helps better justify why the proposed technique is superior over existing multi-/single-object tracking algorithms.

(2) The presentation is also not very easy-to-understand. In particular, the colorful circles are not easy to interpret. Both figure 1 and 2 lacks sufficient clarifications about the patterns used in the figure. What do the colorful circles exactly refer to? Text spotting results in bounding boxes or masks? What are associated? The current presentation style requires better clarification.

(3) It is not clear to me what new insights are contributed in this paper. All the employed technologies are well-known techniques. In particular, I suggest the authors to explicitly state the novel aspects of their approach and how it differs from simply combining existing techniques. How does this ensemble of existing techniques introduce new design ideas in the community? In addition, I do not quite buy the claim of "global" in this paper, please better clarify what they mean by "global" and how their approach achieves this, given the limited number of frames considered. Lastly, Please provide a more comprehensive and informative justification for using Wasserstein distances specifically, perhaps by comparing it to other distance metrics they considered.

(4) Lastly, the experiments seem not very convincing as well. The compared methods look kind of old. Not many more recent approaches are discussed and compared in the results. This limits the contribution of this paper.  For the compared methods, more recent related literature deserves a comparison or at least discussion. This includes but not limited to [5][6].

I would like to see the authors' response before making my final decision.

[1] Xu, Yuanyou, Zongxin Yang, and Yi Yang. "Integrating boxes and masks: A multi-object framework for unified visual tracking and segmentation." Proceedings of the IEEE/CVF International Conference on Computer Vision. 2023.
[2] Shim, Kyujin, et al. "Adaptrack: Adaptive Thresholding-Based Matching for Multi-Object Tracking." 2024 IEEE International Conference on Image Processing (ICIP). IEEE, 2024.
[3] Cao, J., et al. "Observation-centric sort: Rethinking sort for robust multi-object tracking. arXiv 2022." arXiv preprint arXiv:2203.14360.
[4] Cheng, Ho Kei, and Alexander G. Schwing. "Xmem: Long-term video object segmentation with an atkinson-shiffrin memory model." European Conference on Computer Vision. Cham: Springer Nature Switzerland, 2022.
[5] Wu, Weijia, et al. "End-to-end video text spotting with transformer." International Journal of Computer Vision 132.9 (2024): 4019-4035.
[6] Wu, Weijia, et al. "DSText V2: A comprehensive video text spotting dataset for dense and small text." Pattern Recognition 149 (2024): 110177.

---

> ### Author Response · Authors · 2024-11-20
>
> ## Why not simply follow MOT?
> As SOT is not suitable for VTS, we here only discuss MOT.
> ### The differences between MOT and VTS:
> This paper focuses on the notion that while following the established pipeline in MOT has proven useful in previous studies on VTS, there are unique aspects of video text scenes that previous research has overlooked. These differences, compared to objects in MOT or SOT (such as pedestrians, vehicles, or general objects), are essential for enhancing VTS performance.
> As discussed in lines 037-046, two key distinctions arise:
> (a) The visual appearance of texts remains more stable in VTS.
> (b) While the shapes of texts are more stable, their motion can be quite dramatic.
> Texts do not experience limb deformation; instead, most deformations are due to perspective changes, with only a few instances of occlusion. As illustrated in Figure 4, a significant portion of texts moves so quickly that some have no overlapping area even between two consecutive frames.
> ### Mainstream MOT Pipelines and the Introduction of Global and Morphological Information in VTS
> A typical MOT algorithm considers both visual and positional information, or sometimes focuses solely on positional information (e.g., OC-SORT, ByteTrack). These algorithms often rely on features from the most recent frame, occasionally incorporating an updating scheme to maintain implicit attention to earlier frames. This approach is understandable, as visual features in MOT scenarios can be unstable over longer periods. For instance, a pedestrian may turn around between frames. Thus, it makes sense to prioritize the most recent features in MOT.
> In contrast, scene texts themselves do not change; any deformation is typically caused by camera motion. Introducing global information can effectively address issues such as motion blur in recent frames.
> Furthermore, in MOT, positional associations are primarily represented as IoU scores. This limitation arises because MOT datasets use bounding boxes defined by only two points (e.g., tlbr), which lack morphological information. Additionally, the frequent changes in limb position and pose further diminish the contribution of morphological details to the tracking process.
> This situation contrasts sharply with VTS. The shapes of scene texts do not change unless they are rarely occluded. This stability is why we can effectively utilize morphological information in VTS.
> ## Better presentation about figures.
> As indicated in the legends of Figure 1 and Figure 2, the colorful circles represent the embeddings used for association, which are extracted using RotatedRoIAlign based on the detection results (see lines 155-158). The detection results are represented as 4-point coordinate bounding boxes (see line 157). We perform the association between texts in Frame t and those in the global pool (see lines 158-161) to build trajectories. We will include this information in the figure captions for better clarification.
> ## New insights
> As discussed above, this paper introduces global matching for VTS while considering morphological information, given the unique features of VTS compared to MOT. We aim to inspire future research in this field to not only adopt new methods from MOT but also to focus more on the distinct characteristics of VTS. Furthermore, to our knowledge, there are no existing methods that use Wasserstein distance as a matching score.
> When we refer to "global," we do not mean the entirety of a video; instead, it pertains to the lifespan of a text instance. Our focus is not on comprehensively understanding the entire video but specifically on VTS. It is unnecessary to maintain tracking and embeddings for a text that is permanently out of view. As shown in Table 3, we conducted ablation studies on the sliding window size and found no performance gain when further increasing the window size, but it does increase computational cost. Theoretically, we could perform associations across a whole video, but the associated time cost would be prohibitive.
> Given the fact that previous studies in VTS do not focus much on designing morphological associations, we compare the Gaussion Wasserstein distance with L1/2 distance, the results are as follows:
> | Methods                     | MOTA ↑ | MOTP ↑  | IDF1 ↑ | MM ↑   | ML ↓   |
> |-----------------------------|--------|---------|--------|--------|--------|
> | Ours *w.* L1                | 55.5   | **77.5**| 70.3   | 48.3   | 27.9   |
> | Ours *w.* L2                | 55.8   | 77.4    | 70.2   | 49.7   | 27.6   |
> | Ours *w.* Wasserstein       | **56.0**| 77.4   | **70.5**   | **49.7**| **27.3**   |
>
> ## Comparison with recent approaches
> Since recent methods often use various additional training data (e.g., DSText, ArtVideo, BOVText, etc.), making direct comparisons can be unfair. In our paper, we will discuss these methods, include their results, and clearly indicate which ones use additional data for training.

---

> > ### Comment · Reviewer_1EQ7 · 2024-11-21
> >
> > I very much appreciate the authors' comprehensive response, but some of my concerns remain. For example, when talking about the relations between MOT and the VTS, the presented explanations are not very convincing. I understand that VTS may have unique challenges, but I found the answers do not really exclude MOT from the VTS problem. It appears to me that the VTS is a sub-problem of MOT. Also, the discussion that mentions that MOT favors recent features is not very true to me. There are studies focusing on long-term tracking scenarios [1]. I believe this paper requires a more in-depth discussion about the differences between MOT and VTS. Also, the explanation of the term "global" is also not convincing. The authors explain that it "pertains to the lifespan of a text instance". This makes me think that this has a similar effect with long-term tracking approaches. I do not believe "global" is a proper term to summarize the proposed approach. Overall, although this paper introduces an interesting topic, I believe it needs some clearer discussions and more rational justifications. I therefore tend to maintain my original rating.
> >
> > [1] Qin, Zheng, et al. "Motiontrack: Learning robust short-term and long-term motions for multi-object tracking." Proceedings of the IEEE/CVF conference on computer vision and pattern recognition. 2023.

---

> ### Author Response · Authors · 2024-11-21
>
> Thanks for you reply!
>
> I believe we should distinguish between the ideal definition of MOT and the existing methods within this field. Theoretically, VTS can be considered a sub-area of MOT, and I agree with that perspective. However, due to the limited distribution of current MOT datasets, existing methods have significant limitations and are not sufficiently generalized for direct application to VTS.
>
> Most commonly used MOT datasets, such as MOT17, MOT20, TAO, and DanceTrack, primarily feature fixed-camera videos. Although some less mainstream datasets, like KITTI, include camera motion, the movements are generally not rapid. This differs significantly from the motion patterns of texts in VTS, where most movement is caused by camera motion, often resulting in irregular trajectories. In contrast, objects in existing MOT datasets tend to move themselves, albeit more slowly, and are affected by changes in pose.
>
> The MotionTrack method you mentioned aligns with our earlier comment that "visual features in MOT scenarios can be unstable over longer periods." As a result, it seeks solutions based on long-range motion (something like a learnable Kalman Filter often applied in MOT), which tends to be more regular in existing MOT datasets, rather than continuing to rely on visual features.
> ### Summary:
>
> 1. The challenges in VTS and MOT differ significantly:
>
> | Tasks                  | VTS                      | MOT                      |
> |------------------------|-------------------------|--------------------------|
> | Motion             | Irregular, dramatic      | Regular, slow            |
> | Shape            | Stable                   | Unstable                 |
> | Coordinates              | 4-point (or with angle)  | 2-point                  |
> | Occlusion          | Occasionally             | Often                    |
> | Visual Appearance  | Stable                   | Unstable                 |
>
> Consequences:
> - Motion Patterns: MOT heavily relies on positional association. Most MOT methods prioritize positional association scores over visual scores.
> - Shape and Coordinate Patterns: MOT methods do not leverage morphological information.
> - Occlusion and Visual Appearance: MOT often employs motion prediction techniques (e.g., Kalman Filter) to estimate movement during occlusions or relies on visual features from the last observed frame. In contrast, VTS can rely on global visual features or shape similarity. Additionally, the ablation study in Table 4 demonstrates that GLOMA relies more on visual features than on positional and morphological information. This contrasts with MOT methods, which typically depend more on motion cues.
>
> 2. VTT/VTS is only the name of this field. It can be thought of as multi-text tracking. However, this similarity in definition does not imply that current MOT methods are suitable for VTT/VTS. The key point is not whether VTT/VTS is a sub-field of MOT, but rather that current MOT methods do not adequately address the requirements of VTT/VTS.

---

> > ### Comment · Reviewer_1EQ7 · 2024-11-21
> >
> > Thanks for the authors' additional explanations. Based on the new feedback, I have the following comments.
> > (1) The more explanations are very much appreciated, but I think my point was misunderstood. In fact, I am not against the VTS task. I agree that it has its unique challenges that the current MOT approaches cannot deal well with. However, this does not mean that the current MOT methods are not worth comparing. My point was that the MOT approaches can be applied no matter how bad they are, and they can be compared to further highlight the paper's strengths and contributions. I think this is a missing puzzle in the current manuscript. Authors seem to presume that all readers should accept the fact that the most powerful MOT methods also fail to achieve this task without clear evidence.
> > (2) The statement about MOT: "visual features in MOT scenarios can be unstable over longer periods." has been repeated several times. I admit that this could be true in many scenarios, but, based on my years of study on tracking, I personally do not think this is always true. Firstly, the tracked targets like vehicles in existing datasets are not always deformable. Secondly, studies, including the one I referenced, introduced techniques (like memory mechanism [2]) to deal with instability in the long term, which could deal with unstable visual features to some extent, and the motion information is not the only cue we relied on. Nevertheless, this statement is not fully justified in the experiment.
> >
> > Anyway, I agree that this paper has contributed promisingly to the VTS task, but I believe some better clarifications and more comprehensive discussions (together with the experiments) are needed. The authors did not push my swing toward acceptance at the end. I will leave the final decision to the area chair.
> >
> > [2] Cheng, Ho Kei, and Alexander G. Schwing. "Xmem: Long-term video object segmentation with an atkinson-shiffrin memory model." European Conference on Computer Vision. Cham: Springer Nature Switzerland, 2022.

---

> > > ### Author Response · Authors · 2024-11-26
> > >
> > > Per the reviewer's request, we conducted preliminary explorations on applying SoTA MOT methods to VTS. We selected OC-SORT due to its strong performance in MOT and the availability of various variants. Since it is association methods, no model training is required. The results on the ICDAR15 video dataset are as follows:
> > >
> > > | Methods                  | MOTA ↑ | MOTP ↑ | IDF1 ↑ | MM ↑ | ML ↓ |
> > > |-------------------------|--------|--------|--------|------|------|
> > > | Ours w. OC-SORT (*default*)     | 4.9    | 76.7   | 18.1   | 1.6  | 91.8 |
> > > | Ours w. OC-SORT (*better*)      | 12.3   | 76.7   | 23.0   | 2.9  | 90.6 |
> > > | Ours w. (OC-SORT w. Byte)       | 12.3   | 76.7   | 23.0   | 2.9  | 90.6 |
> > > | Ours w. W_t             | 43.2   | 77.8   | 50.2   | 33.0 | 40.2 |
> > > | Ours                    | 56.0   | 77.4   | 70.5   | 49.7 | 27.3 |
> > >
> > > The last two entries in this table correspond to those in Table 4 of the paper. The term OC-SORT (default) refers to the use of the default hyperparameters from the official OC-SORT code repository, while OC-SORT (better) indicates that we adjusted the IoU threshold from 0.3 to 0.01 and reduced $\Delta t$ from 3 to 1. OC-SORT w. Byte refers to the combination of OC-SORT with ByteTrack; however, we did not observe any performance differences.
> > >
> > > Our results demonstrate that lowering the IoU threshold and $\Delta t$ improves performance. Also, reducing $\Delta t$, which decreases the reliance on historical motions for predicting future coordinates, proves beneficial. This suggests that the motions in VTS are often dramatic, leading to less reliable motion cues. Thus, it is reasonable to rely more on other features (e.g., vision, shape). Additionally, for a fair comparison, when relying solely on coordinate-based associations, our proposed methods (4th line) significantly outperform OC-SORT in VTS.
> > >
> > > Considering that this paper is focused on VTS and recognizing the significant gap between MOT methods and VTS methods on VTS benchmarks, we think it might be better to include this experiment in the appendix. However, we would appreciate your feedback on this.

---

> > > > ### Comment · Reviewer_1EQ7 · 2024-11-27
> > > >
> > > > Thanks to the authors for their hard efforts in addressing my concerns at the last minute. I am satisfied with the further evidence to highlight their contributions and would like to raise my rating at the end. Though, the paper needs to be revised carefully and try to emphasize their contributions more clearly.

---

### Official Review · Reviewer_JtLg · 2024-10-31

**Soundness:** 2
**Presentation:** 2
**Contribution:** 2
**Rating:** 5
**Confidence:** 4

**Summary:**

Summary
The article presents a novel method for video text spotting (VTS) that leverages global associations and morphological information, called GLOMA. This approach primarily addresses the limitations of current VTS methods, which typically rely on local associations and IoU-based distance metrics, often ignoring rich temporal and morphological cues. The main contributions of GLOMA include: a Transformer-based global tracking method, a position association method based on the Wasserstein distance, and extensive experiments demonstrating state-of-the-art performance across multiple benchmark datasets.

**Strengths:**

Strengths
1. GLOMA makes a significant contribution to the field by utilizing global associations through the Transformer architecture and integrating morphological information with the Gaussian Wasserstein distance. This effectively addresses the limitations of previous methods that primarily relied on local pairwise comparisons.
2. The paper demonstrates significant performance improvements across multiple datasets. For example, the MOTA on the ICDAR2015 video dataset improved by 4.6 absolute points, and compared to the previous best Transformer-based method, MOTA improved by 8.3 points. These results strongly validate the effectiveness of the GLOMA method.
3. The authors conducted an in-depth ablation study to examine the impact of key components such as sliding window size, distance measurement, attention layers, and semantic embeddings. This provides valuable insights for model design.
4. The paper is well-written, with clear explanations of the method, including architecture, training process, and inference process. Figures and tables effectively illustrate key concepts and results.

**Weaknesses:**

Weaknesses
Although the experimental results are encouraging, there are still several key issues that need further clarification:
1. In the abstract, the sentence “ignoring the abundant temporal information and the morphological characteristics in VTS” conveys the idea of neglecting temporal information and morphological features, but it does not sufficiently emphasize the importance of these features for the video text tracking (VTS) task. It is suggested to more explicitly highlight the core role of temporal information and morphological features in VTS to enhance reader understanding.
2. Although the abstract mentions the "Gaussian Wasserstein distance," it does not explain its specific advantages and role in VTS in detail. Readers may want more information on why this method is more effective than traditional IoU distance. Similarly, the innovations of GLOMA could be slightly expanded to show its contributions to solving existing problems.
3. The transitions between different sections in the introduction are not very smooth. For example, the description of existing research methods directly jumps to the introduction of the new model GLOMA, lacking transitional paragraphs to clarify why a new model is needed. It is suggested to add some transition sentences to better guide readers from existing work to the proposal of the new model.
4. Figure 1 does not clearly indicate the specific types of failure situations (such as target loss or target error) before directly introducing global associations to reduce the probability of failure. Perhaps the phrase "To solve the problems" could be removed, and the role of global associations could be directly stated. Additionally, other types of failure situations could be added to enhance persuasiveness and comprehensiveness of the explanation.
5. In section 3.1 Overview, it is mentioned that GLOMA has three parallel modules: detection module, recognition module, and tracking module. However, these modules are not clearly represented in Figure 2, and the aesthetic and logical clarity of the illustration is lacking. It is recommended to use more distinct dividing lines to separate the modules and add text descriptions in the figure, especially in the right-side process section, to enhance overall visualization.
6. The statement "Wasserstein distance can capture both location similarity and morphological similarity" theoretically makes sense, but it lacks detailed explanation on how it specifically captures morphological similarity. It is suggested to add an explanation of how the Wasserstein distance is used to measure morphological features, helping readers understand its advantages over traditional IoU.
7. In equation (8), it is mentioned that α is a hyperparameter, but the text does not detail how to choose this hyperparameter and its impact on model performance. Moreover, the form and role of the function f are not thoroughly discussed. It is recommended to provide details on the hyperparameter selection process, especially the normalization process of f and its impact on the results.
8. Although the text provides a general framework for embedding pool and trajectory association, it lacks detailed explanations about the embedding computation and updating mechanism. For example, how are embeddings updated? How is the length of the sliding window selected? Is there a trajectory drift problem? It is suggested to further elaborate on the embedding update strategy and trajectory construction details during inference.

**Questions:**

Please refer to my previous comments.

---

> ### Author Response · Authors · 2024-11-20
>
> ## Highlight temporal information and morphological features in VTS.
> We agree with your suggestions and we will add some illustrations in Figure 1.
> ## Gaussian Wasserstein distance
> As illustrated in Figure 3, we present three cases where Gaussian Wasserstein distance outperforms the IoU score due to its consideration of morphological information. This method can be extended to other MOT scenarios if objects are represented as rotated boxes. However, currently mainstream MOT datasets do not take the angle of bounding boxes into account.
> ## Equation 8
> The normalization term helps mitigate the impact of the absolute values of the boxes' width and height. We can do a simple math derivation here. Assume that $\\theta_1$ is $0$ and $\\theta_2$ is $\\pi/2$, we can get:
> 1. Firstly:
>    - $\\sigma_1=\\begin{pmatrix}\\frac{w_1}{2}&0\\\\0&\\frac{h_1}{2}\\end{pmatrix}$. Then, $\\sigma_1^{1/2}=\\begin{pmatrix}\\sqrt{\\frac{w_1}{2}}&0\\\\0&\\sqrt{\\frac{h_1}{2}}\\end{pmatrix}$.
> 2. Secondly:
>    - $\\sigma_2=\\begin{pmatrix}\\frac{h_2}{2}&0\\\\0&\\frac{w_2}{2}\\end{pmatrix}$. And $\\sigma_2^{1/2}=\\begin{pmatrix}\\sqrt{\\frac{h_2}{2}}&0\\\\0&\\sqrt{\\frac{w_2}{2}}\\end{pmatrix}$.
> 3. Next:
>    - $\\sigma_1^{1/2}\\sigma_2\\sigma_1^{1/2}=\\begin{pmatrix}\\sqrt{\\frac{w_1}{2}}&0\\\\0&\\sqrt{\\frac{h_1}{2}}\\end{pmatrix}\\begin{pmatrix}\\frac{h_2}{2}&0\\\\0&\\frac{w_2}{2}\\end{pmatrix}\\begin{pmatrix}\\sqrt{\\frac{w_1}{2}}&0\\\\0&\\sqrt{\\frac{h_1}{2}}\\end{pmatrix}=\\begin{pmatrix}\\frac{w_1h_2}{4}&0\\\\0&\\frac{w_2h_1}{4}\\end{pmatrix}$.
> 4. Then:
>    - $(\\sigma_1^{1/2}\\sigma_2\\sigma_1^{1/2})^{1/2}=\\begin{pmatrix}\\sqrt{\\frac{w_1h_2}{4}}&0\\\\0&\\sqrt{\\frac{w_2h_1}{4}}\\end{pmatrix}$.
> 5. After that:
>    - $\\sigma_1+\\sigma_2 - 2(\\sigma_1^{1/2}\\sigma_2\\sigma_1^{1/2})^{1/2}=\\begin{pmatrix}\\frac{w_1}{2}+\\frac{h_2}{2}-2\\sqrt{\\frac{w_1h_2}{4}}&0\\\\0&\\frac{w_2}{2}+\\frac{h_1}{2}-2\\sqrt{\\frac{w_2h_1}{4}}\\end{pmatrix}=\\begin{pmatrix}(\\sqrt{\\frac{w_1}{2}}-\\sqrt{\\frac{h_2}{2}})^2&0\\\\0&(\\sqrt{\\frac{w_2}{2}}-\\sqrt{\\frac{h_1}{2}})^2\\end{pmatrix}$.
> 6. Then:
>    - $\\text{Tr}(\\sigma_1+\\sigma_2 - 2(\\sigma_1^{1/2}\\sigma_2\\sigma_1^{1/2})^{1/2})=(\\sqrt{\\frac{w_1}{2}}-\\sqrt{\\frac{h_2}{2}})^2+(\\sqrt{\\frac{w_2}{2}}-\\sqrt{\\frac{h_1}{2}})^2$.
> 7. According to $d^2=\\|\\mu_1 - \\mu_2\\|^2+\\text{Tr}(\\sigma_1+\\sigma_2 - 2(\\sigma_1^{1/2}\\sigma_2\\sigma_1^{1/2})^{1/2})$, assuming $\\mu_1=\\mu_2=(0,0)$, we have:
>    - $d^2=(\\sqrt{\\frac{w_1}{2}}-\\sqrt{\\frac{h_2}{2}})^2+(\\sqrt{\\frac{w_2}{2}}-\\sqrt{\\frac{h_1}{2}})^2$. Expanding it, we get $d^2=\\frac{w_1}{2}- \\sqrt{w_1h_2}+\\frac{h_2}{2}+\\frac{w_2}{2}-\\sqrt{w_2h_1}+\\frac{h_1}{2}=\\frac{w_1 + w_2+h_1 + h_2}{2}-\\sqrt{w_1h_2}-\\sqrt{w_2h_1}$.
> 8. Now, for calculating $W$. We know that $W(b_1,b_2)=1-\\frac{\\alpha d}{f(\\sigma_1,\\sigma_2)}$, where $f(\\sigma_1,\\sigma_2)=(\\text{Tr}(\\sigma_1\\sigma_2))^{1/4}$.
>    - First, find $\\text{Tr}(\\sigma_1\\sigma_2)$. Given $\\sigma_1=\\begin{pmatrix}\\frac{w_1}{2}&0\\\\0&\\frac{h_1}{2}\\end{pmatrix}$ and $\\sigma_2=\\begin{pmatrix}\\frac{h_2}{2}&0\\\\0&\\frac{w_2}{2}\\end{pmatrix}$, we have $\\sigma_1\\sigma_2=\\begin{pmatrix}\\frac{w_1h_2}{4}&0\\\\0&\\frac{w_2h_1}{4}\\end{pmatrix}$. So, $\\text{Tr}(\\sigma_1\\sigma_2)=\\frac{w_1h_2 + w_2h_1}{4}$.
>    - Then, $f(\\sigma_1,\\sigma_2)=(\\frac{w_1h_2 + w_2h_1}{4})^{1/4}$.
>    - And $d=\\sqrt{\\frac{w_1 + w_2+h_1 + h_2}{2}-\\sqrt{w_1h_2}-\\sqrt{w_2h_1}}$.
>    - Finally, substituting these into the formula for $W$, we get:
>    - $W(b_1,b_2)=1-\\frac{\\alpha\\sqrt{\\frac{w_1 + w_2+h_1 + h_2}{2}-\\sqrt{w_1h_2}-\\sqrt{w_2h_1}}}{(\\frac{w_1h_2 + w_2h_1}{4})^{1/4}}$.
>    - If we set $w_1=w_2=k*h_1=k*h_2$, we have:
>    - $W(b_1,b_2)=1-\\frac{\\alpha\\vert\\sqrt{k} - 1\\vert}{\\left(\\frac{k}{2}\\right)^{1/4}}$
>
> So that $W(b_1,b_2)$ is not influenced by the absolute values of w, h.
> α is expected to rescale the Wasserstein distance score into a suitable range. We set it as 1.0.
> ## Embedding computation and updating mechanism
> As discussed in lines 281-297, during inference, we maintain an embedding pool to store embeddings of text instances within a sliding window. For each new frame, we attempt to associate its text instances with those in the global pool. If a match is found, we update the track. If a new text instance doesn't match any in the pool, it indicates the start of a new trajectory. Table 3 presents ablation studies on the sliding window size. Since GLOMA is a tracking-by-detection process in VTS, it experiences minimal tracker drift issues.
> ## Writing issues
> We will add a transitional paragraph to clarify the necessity of GLOMA, emphasizing both global and morphological information. Additionally, we will refine Figure 1 to clearly illustrate more failure scenarios and enhance the clarity of the figure captions. Furthermore, we will include the recognition module in Figure 2, which was initially omitted for clarity.

---

### Official Review · Reviewer_qTr3 · 2024-11-01

**Soundness:** 3
**Presentation:** 3
**Contribution:** 3
**Rating:** 8
**Confidence:** 4

**Summary:**

This manuscript introduces GLOMA, a Transformer-based global tracking approach that frames the tracking challenge as a problem of global associations. It employs the Gaussian Wasserstein distance to facilitate morphological correlation across frames. The efficacy of GLOMA is demonstrated through experimental results on renowned benchmarks, including the ICDAR 2015 video dataset. The contributions of this manuscript are:
1. A Transformer-based global tracking method for VTS that associates frames globally.
2. A Wasserstein distance-based method for measuring morphological similarity.
3. Extensive experiments on public datasets, demonstrating state-of-the-art performance.

**Strengths:**

**Global Association Approach**: The paper introduces a Transformer-based global tracking method that simultaneously associates multiple frames, effectively utilizing global context information in video sequences.

**Gaussian Wasserstein Distance**:
The use of Wasserstein distance for positional association considers both location and morphological features, enhancing the accuracy of matching, especially in scenarios with fast movement or interference.

**Performance Improvements**:
Achieving a 56.0 MOTA on the ICDAR2015 video dataset, with a 4.6 absolute improvement over the previous state-of-the-art method and an 8.3 MOTA improvement over the previous Transformer-based method, demonstrating significant performance gains.

**Extensive Experimental Validation**:
The paper conducts extensive experiments on public datasets, validating the effectiveness of the GLOMA method, which proves its practicality and efficacy in the field of video text recognition.

**Weaknesses:**

**The Choice of the Text Detector/Spotter**:
The paper discusses the use of a 4-point coordinate prediction head for text detection within the GLOMA framework. However, it does not incorporate state-of-the-art text detectors or text spotters capable of detecting text of arbitrary shapes, such as DBNet++ and Mask TextSpotter v3. It would be beneficial to understand why these advanced detectors or spotters were not utilized in the proposed method. Additionally, the paper should ideally include a discussion on existing scene text spotting methods as part of its related work, given that these are foundational to the field of video text spotting.

**The Use of the Synthetic Data**:
Regarding the use of synthetic data for training, as demonstrated in TransDETR, synthetic data can significantly enhance model performance when used as pretraining data. For instance, TransDETR achieved a 51.8 MOTA score on the ICDAR 2015 video dataset with synthetic data, compared to a 47.7 MOTA score without it. It would be insightful to know why the proposed GLOMA method did not leverage synthetic data for training purposes. Addressing these points could provide a more comprehensive understanding of the design choices made in developing the GLOMA framework and how it compares to other leading approaches in the field.

**Questions:**

Please give the responses to the concerns in the Weaknesses.

---

> ### Author Response · Authors · 2024-11-20
>
> ## The Use of Synthetic Data
> At a high level, this paper aims to explore effective methodologies for video text spotting rather than merely striving for top performance, in other words, this is a method paper other than a data paper. Therefore, ensuring a fair comparison is crucial, as most methods in this field follow a similar pipeline: they pretrain on the COCO-Text dataset and then fine-tune on downstream datasets. While some studies utilize different datasets (e.g., BOVText, DSText, ArtVideo, etc.), our approach does not incorporate additional data in line with our main focus. We think ensuring fair comparison is more important, even though scaling the data often yields benefits.
> ## The Choice of the Text Detector/Spotter
> Furthermore, this paper aims to delve into end-to-end video text spotting, which necessitates consideration of performance across detection, tracking, and spotting tasks, rather than focusing solely on one or two aspects. We conducted early explorations using advanced architectures of text spotters, training from scratch while ensuring fairness by not using pretrained model weights. However, we found that the architectures of these text spotters were not well-suited for unified tasks.
> We appreciate your suggestion to include some scene text spotting methods in our related works section. We will make the necessary revisions to enhance the clarity and depth of our paper.

---

### Official Review · Reviewer_Qrc3 · 2024-11-02

**Soundness:** 4
**Presentation:** 3
**Contribution:** 3
**Rating:** 8
**Confidence:** 3

**Summary:**

This article proposes a more effective tracking method for video text based on the characteristics of the text. In specific, The paper introduce GLOMA to model the tracking problem as global associations and utilize the Gaussian Wasserstein distance to guide the morphological correlation between frames. GLOMA achieves state-of-the-art performance on multiple video text recognition benchmarks.

**Strengths:**

1. The article is clearly written and easy to understand.

2. The motivation is strong, Wasserstein distance is indeed a better measurement for video text association.

3. Extensive experiments demonstrate the effectiveness of the proposed new association method.

4. The proposed method demonstrates improved performance compared to several SOTA methods across multiple benchmarks.

**Weaknesses:**

1. Compared to TransDETR, which uses learnable query for object tracking, what is the advantage of the proposed association method.

**Questions:**

See weaknesses

---

> ### Author Response · Authors · 2024-11-20
>
> The proposed association method integrates global visual information, positional information, and morphological details, which are crucial in VTS scenarios. In contrast, TransDETR primarily adheres to the MOT pipeline, focusing on local matching and IoU-based positional association. Our approach places greater emphasis on the inherent characteristics of scene texts in videos.

---

### Meta-Review · Area_Chair_bzyA · 2024-12-20

**Metareview:**

This paper addresses video text spotting (VTS) problem and proposes a method called GLOMA. The proposed approach shifts from local to global associations in tracking, while also incorporating morphological information between frames. GLOMA directly addresses the shortcomings of existing techniques that primarily rely on IoU-based metrics and localized associations, often overlooking temporal and shape-related details. The key contributions of this work include a Transformer-based global tracking framework, a method for establishing positional links using a Gaussian Wasserstein distance, and comprehensive experiments showcasing superior performance on standard benchmark datasets.

Reviewers variously praised the article for its clarity in writing and presentation which thus effectively communicate its main contributions, for the comprehensive experimental evaluation, for the impressive performance gains especially on the ICDAR2015 dataset, and for the novel incorporation of the Gaussian Wasserstein distance which facilitates associations between observations differing in both spatial and morphological attributes.

The weak points identified by reviewers mostly concentrate on questions about novelty, questions regarding the choice to omit the use of synthetic data and state-of-the-art components for text detection, and some need for clarifications. The authors responded adequately to these concerns in rebuttal and the decision is to Accept.

**Additional Comments On Reviewer Discussion:**

The authors responded adequately to all reviewer concerns in rebuttal, and the general consensus was very positive due to the novelty of incorporating the Gaussian Wasserstein distance and the excellent performance improvements over the state-of-the-art.

---

### Decision · Program_Chairs · 2025-01-22

Accept (Poster)